# Austerity Measures and Underfunding of the Spanish Health System during the COVID-19 Pandemic—Perception of Healthcare Staff in Spain

**DOI:** 10.3390/ijerph20032594

**Published:** 2023-01-31

**Authors:** Laura Otero-García, José Tomás Mateos, Alexo Esperato, Laia Llubes-Arrià, Vanesa Regulez-Campo, Carles Muntaner, Helena Legido-Quigley

**Affiliations:** 1CIBER Epidemiology and Public Health (CIBERESP-ISCIII), 28029 Madrid, Spain; 2Nursing Department, Faculty of Medicine, Universidad Autónoma de Madrid, 28029 Madrid, Spain; 3Department of Nursing and Physiotherapy, Faculty of Nursing and Physiotherapy, University of Lleida, 25199 Lleida, Spain; 4Healthcare Research Group (GRECS), Institute of Biomedical Research in Lleida (IRB Lleida), 25198 Lleida, Spain; 5Asian Development Bank, Manila 1554, Philippines; 6Osakidetza, Nursing Teaching Unit, Cruces University Hospital, 48903 Baracaldo, Spain; 7Faculty of Nursing, Dalla Lana School of Public Health, University of Toronto, Toronto, ON M5T 3M7, Canada; 8Department of Global Health and Development, London School of Hygiene and Tropical Medicine, London WC1E 7HT, UK; 9Saw Swee Hock School of Public Health, National University of Singapore, Singapore 117549, Singapore

**Keywords:** COVID-19, health policy, healthcare financing, qualitative research, Spain

## Abstract

Insufficient pandemic preparedness and underfunding of human and economic resources have conditioned the response to COVID-19 in Spain. This underfunding has continued since the austerity measures introduced during the 2008 financial crisis. This study aims to understand the perceptions of healthcare staff in Spain on the relationship between the funding of the health system and its capacity to respond to the COVID-19 pandemic. To this end, we carried out a thematic content analysis, based on 79 online semi-structured interviews with healthcare staff across the regions most affected by the COVID-19 first wave. Participants reported a lack of material resources, which had compromised the capacity of the health system before the pandemic. The lack of human resources was to be addressed by staff reorganisation, such as reinforcing hospital units to the detriment of primary health care. Staff shortages continued straining the COVID-19 response, even after material scarcities were later partially alleviated. Personnel shortages need to be adequately addressed in order to adequately respond to future health crises.

## 1. Introduction

The rapid spread of SARS-CoV-2 tested the capacity of the Spanish health system to meet increased demand for healthcare [1]. The availability of resources, both human and material, is a fundamental component in the resilience of health systems [2]. In the case of Spain, the lack of human resources, together with the scarcity of material resources and infrastructure—especially in public health services, primary health care, and intensive care units (ICUs) lead, in some regions, to the collapse of all levels of care, especially during the first weeks of the pandemic [3].

On 14 March 2020, the Spanish government implemented RD 463/2020, which declared a state of alarm for the management of the COVID-19 pandemic [4]. This response was accompanied by an increase in funding and a reorganisation of the health system [1]. The Ministry of Health assumed sole responsibility for managing the additional COVID-19 response resources, while the regional governments retained management of the health services. In addition, the most urgent hospital services were prioritised, thus mobilising professionals from primary health care centres to hospitals and emergency facilities, which led to staff shortages in essential services [5].

Numerous studies show that austerity measures introduced after the 2008 financial crisis, such as the reduction of public spending on healthcare, have a direct impact on healthcare provision [6,7] and the daily practice of healthcare staff [8,9,10]. In Spain, public spending on healthcare fell indeed by 1.4% in the period from 2008 to 2013 [11]. Although it increased from 2013 onwards, by 2019 (the year prior to the health crisis), healthcare investments were still below the European Union and OECD averages, without having recovered the levels prior to the financial crisis [12,13].

Healthcare workers are vital to ensure continued services during crises [14]. During the COVID-19 pandemic, previously precarious working conditions [15,16,17,18] were compounded by high levels of stress and uncertainty. Therefore, understanding how the health system’s baseline situation is relevant to a correct assessment of the COVID-19 response [19,20]. Therefore, this study aims to describe the perceptions of healthcare staff on the relationship between the resources of the Spanish National Health System, and its capacity to respond to the COVID-19 pandemic during its first wave.

## 2. Materials and Methods

### 2.1. Study Design

This study aims to elicit healthcare worker perceptions on the relationship between the National Health Service’s baseline resources, and its COVID-19 response [21]. These data thus contribute to our understanding of COVID-19 response in Spain, and is part of the research project “Health Crisis Management during COVID-19”.

Through a descriptive phenomenological approach [22], we conducted semi-structured interviews with 79 participants (see Table 1). To this end, we identified professionals who worked in the healthcare services of the autonomous communities most affected by the first COVID-19 wave (Castilla-La Mancha, Castilla y León, Catalonia, Galicia, Madrid and the Basque Country). Adopting a purposive sampling approach, participants were recruited from direct contact and snowballing of professionals, whether they provided care for COVID-19, or other pathologies.

### 2.2. Data Collection

A member of the research team (HLQ) designed a draft of the interview guide based on the “health service delivery”, “health workforce”, and “health systems financing” WHO health system cubes [2]. The rest of the research team contributed added additional questions per local context. The final interview guide included topics related to the management of the COVID-19 pandemic; for this article, the content related to the resources available in the health system was analysed (see Table 2). The fieldwork was carried out by the entire research team between April and June 2020, coinciding with the first wave of COVID-19 cases in Spain. All interviews, which were conducted online, were recorded and lasted on average between 45 and 120 min.

### 2.3. Data Analysis

Each interview was transcribed, and then read by two members of the research team (LOG and JTM), in order to identify patterns (similar and divergent experiences) between the participants’ perception. To systematize the findings, textual fragments were coded in an emergent process, which took into account the pre-established categories according to the WHO health system cubes described above [2].

The coding of textual fragments resulted in two overarching themes of analysis. The first we summarises participant impressions on the impact of austerity measures on COVID-19 response. The second theme explores the management of the health crisis capacity through increased investments or reorganisations of staff and equipment. From each of the main themes, sub-themes were extracted to facilitate understanding of the narratives. Subsequently, each member of the research team carried out a detailed reading of the summarized findings to verify that they represented the interviews conducted. As part of the iterative process, some new categories emerged within each main theme and were agreed upon by the entire team [23].

### 2.4. Rigour

The rigour of the study was ensured by following the Lincoln and Guba [24]’s trustworthiness criteria for qualitative research. The study proceeded in an emergent and iterative process, which allowed continuous adaptation as to deepen the findings. Interview collection and analysis were carried out in parallel, and the researchers were able to introduce changes during the process. Interviews were conducted until, by consensus, the research team considered the information saturation point had been reached. To ensure the credibility and validity of the results, a triangulated analysis was first carried out by two researchers (LOG and JTM), with the support of another researcher (HLQ) in case of discrepancies. In addition, the findings from the first analysis were agreed upon by the entire research team. The sampling of professionals from different regions and health services improved the representativeness of different experiences, and strengthened the credibility of the results. The results section features literal quotes from the participants, as to support the interpretation of the results, and to ensure their confirmability.

### 2.5. Ethical Considerations

Each professional signed an informed consent form stating the objectives of the study, motivation of the interview, and guaranteeing anonymity and confidentiality. To preserve the latter, each interview was coded using the participant’s profession, region, and interview number. The interviews also emphasized that participation was voluntary and, participants could withdraw at any time during the interviews without explanation. The project was approved by the Research Ethics Committee of the University Hospital Arnau de Vilanova of Lleida (CEIC-2278).

## 3. Results

We conducted individual interviews of 79 professionals, 38% of whom were nurses, 24% physicians, 12% ICU nurses, 10% ICU physicians, and 16% were managers and other professionals. The analysis was grouped into two main themes summarizing various subthemes. A summary of the subthemes is shown in Table 3 and Table 4.

### 3.1. Impact of Austerity Measures on the Management of the COVID-19 Crisis

According to the participants, the health system was already under stress before the COVID-19 epidemic. In this regard, they recurrently referred to the impact on the health system of the austerity measures adopted during and after the 2008 financial crisis, and how these measures were not sufficiently mitigated in the years of recovery. They partly blamed this lack of funding on the limited response capacity during the first weeks of COVID-19 expansion, especially in terms of infrastructure and equipment.


*“I obviously understand that all cuts imply a subsequent precariousness and what I have experienced was the closure of beds during the flu season, the closure of beds in summer… I understand this was part of the cuts, and never understood how in the middle of the flu pandemic two years ago the hospital was full but some wards were empty”.*
(ICU physician, Community of Madrid)

Austerity measures after the 2008 economic crisis also resulted in staff reductions and job insecurity for health workers. This had particular influence on key services such as primary health care and public health, which, with insufficient staff and resources, had to cope with demand increases. During the pandemic, lack of staff was alleviated by professionals from other services, additional recruits, and senior students. Even so, personnel shortages were recognised, a situation aggravated by the high number of staff infections.


*“[…] they are teams that are already overloaded from the outset, all of them… In primary care, I think more so than in hospital care. For example, if we go on holiday we are never replaced, so we always have to take on the work of those who leave […] Work overload is the standard”.*
(physician, Galicia)


*“[…] there were times when we even had to pull final year nursing students, because there was no (staff); we also had to take people out of primary care to reinforce hospitals, for example. You are removing one saint to dress another, you are taking people from one place that is necessary to put them in another place that is also necessary. But of course, we have had a (personnel) deficit for years and even more so in nursing homes, because the (personnel) ratios are very low. We have been saying this for a long time and nobody listens. If we had been staffed as before 2008, surely there would have been a different response”.*
(manager, Catalonia)

Participants also reported a shortage of material resources, especially lack of personal protective equipment, which is perceived as a cause of staff infections. In some cases, this shortage was made up for by donations from both the public and private sectors, as well as by reusing of equipment.


*“The lack of material has been managed by continually calling the (nursing) supervisor… threatening to call the press and the unions. Our colleagues have taken out the sewing machine and made caps, homemade masks… We have also received donations from volunteers”.*
(Nurse, Community of Madrid)

Specifically, participants explained the impact of spending cuts on structural, human, and material resources on both ICUs and other services in high demand during the pandemic. On the one hand, high hospital occupancy required closer patient triage, which posed an ethical dilemma to professionals. On the other hand, staff reinforcements, in most cases without specific training on critical care, did not help alleviate ICU saturation.


*“There is a percentage of people who end up in the ICU and, above all due to patient overload and lack of ICU beds, doctors are faced with the fundamental dilemma of having to intubate one person instead of another, and that is the big problem. I believe that yes, cuts in healthcare end up being paid for”.*
(Nurse, Community of Madrid)

In addition to the underfunding from austerity measures, participants also pointed to the outsourcing of public healthcare to private companies as an additional resource constraint. They also explained that the Spanish National Health Services’ shortcomings were aggravated by spending cuts.


*“The cuts after the financial crisis have affected the response to the pandemic. Being a privately managed hospital, it is something that formed part of the policy of cuts and one of the packages that were made was that the management of a hospital that was public was externalised and they have austere management and we already had problems before [the pandemic], especially in terms of human resources”.*
(ICU doctor, Community of Madrid)

### 3.2. Crisis Management and Increased Funding in Response to COVID-19

The participants also stated that the management of the health crisis, without adequate control of investments, was also a major challenge. As an example, they they reported protracted recruitment times. In some cases, the solution was to extend the working hours of health staff. On the other hand, a lack of foresight in equipment procurement impacted its availability and quality, which in many cases was already considered poor.


*“I think that centralised management (of COVID additional resources) made access to resources very difficult, not because there weren’t any, but because at the beginning there was a total lack of control, because people were very nervous. The resources are not all that we would have liked, nor what we would have liked, but in general the staff perceived that the resources were scarce”.*
(manager, Galicia)

In this regard, some participants criticised the transfer of staff between levels of care, especially from primary to hospital care, and the additional resources that were put in place to alleviate the burden of care, such as field hospitals or the use of public spaces as hospital centres.


*“Let’s put it this way, in primary care there has been a significant reduction in staff, forcing them to move to hospitals. On the other hand, at the hospital level, there has been an increase in funding, new or relocated staff who were exclusively dedicated to wards or areas of patients with COVID-19 pathology”.*
(nurse, Catalonia)

The additional resources for COVID-19 response resulted sometimes in the creation of new ICUs in spaces not foreseen for this purpose, with equipment not always adequate and insufficiently trained staff, thus impacting quality of care.


*“I know that in my organisation they set up an ICU, a unit without qualified people, which seems terrible to me, and yes, some Chinese respirator must have arrived, I don’t know how many arrived […] but even so, it has been a bit of a disaster, you can’t set up an ICU when nobody has worked in the ICU… They throw you to the bull, they tell you, there you have a respirator and there you have an ICU patient. That’s how people have been working”.*
(ICU nurse, Basque Country)

Participants indeed perceived a COVID-19 resource increase, particularly for critical care. However, they also explained that the increase was lower than expected, as the conditions in which critical patients were treated were not the usual or necessary ones. They also explained that the resource increases could not always be put to the best use.


*“I think that…there have been resources, money and effort, for example, the field hospital…And where was the money and effort to get EPIS? […] I think that the management in this sense is negative… I don’t know if the military worked for a week to set up the (field) hospital, but it was not used… All that effort and all that money…”.*
(nurse, Castilla y León)

On the other hand, some professionals reported that they perceive a reorganization of services rather than resource increase.


*“We have not seen an increase in funding…We have had little increase in staffing, only one additional nurse per shift, which was not enough. At the beginning, the staff was somewhat reinforced. Now they have moved us back to the original number of staff, even when we are caring for several COVID positive patients”.*
(nurse, Community of Madrid)

## 4. Discussion

In this study, we have collected the perceptions of various health professionals on how the resources available at Spain’s public healthcare facilities conditioned its COVID- 19 response. For most participants, there was a palpable connection between the previous underfunding of the health system and the COVID-19 response deficiencies. Interviewed staff reported both staff and material shortages, especially in some essential services. These findings are consistent with similar studies conducted in Spain and elsewhere during the first months of the pandemic [25,26].

Since the 2008 financial crisis, and its ensuing austerity measures, the Spanish health system has suffered chronic underfunding, which persists to the present day [13]. These measures include lower salaries for public sector professionals, reduced staff recruitment, and budget cuts [27]. Investment in public health services has not reach pre-crisis levels, even during the economic recovery from late 2010s [28,29]. Underfunding has been most significant in essential areas such as primary health care and public health, whose share of public health system funding fell from 14.3% in 2009 to 13.9% in 2018 [5]. The negative impact of the post-financial crisis austerity measures adopted in several European has been widely documented across Europe [7,30], and also in Spain’s health system [31].

The lack of material and human resources is evident in the experience of staff during the first wave of the COVID-19 pandemic, which is the period addressed in this study. Similarly, other studies conducted during the first phase of the pandemic reported a shortage of protective equipment [32,33]. Participants described that the health system, before the COVID-19 crisis, was already weak in its response to seasonal problems such as influenza and some healthcare services could be overwhelmed, which is consistent when compared to epidemiological studies on seasonal influenza responsiveness. [34]. Our findings on the perception of health system funding are in line with other qualitative studies carried out in Spain, where professionals identified the lack of resources, especially human resources, and the weakening of the health system as a major problem [35,36].

This lack of resources puts added pressure on strained healthcare staff, which can ultimately lead to mental health problems [15,37]. In agreement with our findings, other studies have shown how some services are especially vulnerable, as in the case of the ICU, where workers reported mental health problems arising from the conditions under which critically ill patients were treated, particularly during their isolation from family members and healthcare staff [38,39]. Another aspect described by our participants, which is closely related to underfunding, are the ethical challenges that ensure from the inability to adequately treat all patients due to a lack of equipment. This same problem has been described in other studies in Spain and in other countries, especially in those where the health system was less prepared [40].

Although additional funding was made available after pandemic onset, it was reported to be partly ineffective. Participants mainly criticised the lack of foresight, and compared it to the management of the A-influenza pandemic, for which they reported protocols the availability of protocols and resource allocation were in place. This perception is consistent with previous reports that pointed to a lack of preparedness at the European level for a major, unforeseen epidemiological crises [41,42,43].

The results of this study are based on the personal perceptions of participants, which may limit the generalisability of conclusions. Yet, we believe that our sample’s diversity of regions, type of services, and professionals provides sufficient heterogeneity for it to be representative. Moreover, to ensure the rigour of these conclusions, the results were extracted, systematised, and synthesised by two researchers (LOG and JTM) independently and in agreement with the rest of the team.

## 5. Conclusions

From the point of view of healthcare staff, the underfunding of the Spanish healthcare system, which resulted from the austerity measures introduced after the 2008 financial crisis, has reduced its capacity to respond to the COVID-19 pandemic. Although resource scarcity (both human resources and materials) was partially alleviated in the weeks following the first wave, baseline infrastructure and staff shortages have constrained fundamental services for dealing with the pandemic, such as primary health care and public health. Adequate preparedness to future health crises will require adequately resourcing, especially on the human resources front.

## Figures and Tables

**Table 1 ijerph-20-02594-t001:** Territorial and professional distribution of the participants in the interviews.

Autonomous Communities	Physicians	Nurses	ICU Physicians	ICU Nurses	Managers	Others	Total
Castilla y León	3	4		1	1		9
Castilla-La Mancha		2	1				3
Catalonia	9	7	3	4	2	3	28
Madrid	3	7	2	3	2		17
Galicia	2	3			1	2	8
Basque Country	2	7	2	2		1	14
TOTAL	19	30	8	10	6	6	79

**Table 2 ijerph-20-02594-t002:** Semi-structured interview guide.

Objective of the Questions	Questions
Explore organizational and pandemic management.	Could you tell us what your job is like? How do you work as a team? How have you handled the changes in healthcare protocols during the COVID-19 pandemic? Do you believe that the measures adopted by the health service have been adequate? Have they been taken on time? Do you believe that contingency plans were adequate?
Explore effect of austerity measures and increased funding.	How do you think austerity measures following the financial crisis affected COVID-19 response? Have you noticed in your day-to-day work the resource increase for the COVID-19 pandemic?
Explore availability of material resources to address the COVID-19 pandemic.	How were resources managed in your unit? Have you perceived a lack of material resources in your unit? Have you felt protected in your workplace? What would you have needed to feel protected? Have you experienced the lack of PPE, and was it solved? Did you have the necessary drugs available for the care of your patients? Were rapid tests available in your unit, and was the confirmation of positive cases adequate?
Explore human resources situation during the COVID-19 pandemic.	Did your staff increase or decrease? If it increased, indicate by whom and if decreased, indicate to which destinations they were transferred. Were there COVID-19 cases among your staff and if so, what measures have been taken? How were they managed by the organization?

**Table 3 ijerph-20-02594-t003:** Subthemes and representative quotations for the main theme “Impact of austerity measures on the management of the COVID-19 crisis”.

Subtheme	Representative Quotation
Previous system limitations	*I firmly believe that the cuts have had the impact of reducing our response capacity by more than 30 percent. We were coming from a very unfortunate situation. Many professionals had left, there was a lack of generational replacement.*
Underfunding perception	*Yes, indeed, there has been a shortage of personnel, funding and material for many years, and we have clearly seen what has happened with the pandemic now.*
Impact on human resources	*As usual, at certain times there was a need for more personnel and even beds, and there has been reluctance to hire more personnel or to reinforce.*
Impact on material resources	*In the past there were not enough resources, but now the lack of resources has become more evident. I do believe that it has had an influence, undoubtedly, the thing is that before there was not a health emergency, so it was not so evident. But the basic situation was precarious and now, with all this, it has become more evident.*
Privatization of healthcare services	*As it is a privately managed hospital, it was part of the policy of cutbacks and one of the packages that were made was to externalize the management of a hospital that was public and they have an austere management and we already had problems before [the pandemic], especially in terms of human resources.*
Working conditions	*We also, on a personal level, the 37.5 [weekly work hours] are still with them, which makes your stress and your work schedule much higher.*
Primary health care	*The person performs two jobs: his and his colleague’s. That happens with nurses and physicians. They stop having their own patients, their own… and everything. That is, the primary care structure has been broken and they are only dedicated to the assistance and to what they can. Preventive and treetment capacity has been undermined.*
Intensive care unit	*A large number of ICU beds are needed, and since there are none, physicians are faced with the fundamental dilemma of having to intubate one person instead of another, and that is the big problem. I believe that, yes, cuts in health care end up being paid for.*

**Table 4 ijerph-20-02594-t004:** Subthemes and representative quotations for the main theme “Crisis management and increased funding in response to COVID-19”.

Subtheme	Representative Quotation
Management perception	*The crisis has had an influence, but I believe that there has been very bad management, hospital mismanagement, and political mismanagement.*
Staff recruitment	*What is true is that a lot of people had to be recruited. In some hospitals they even had to recruit nursing students because they did not have enough staff.*
Reinforcement infrastructures	*In my hospital, the gymnasium, the cafeteria, the library, the auditorium were fitted out, and there were no beds to fill those spaces. They had to bring them from another hospital in regular conditions.*
Increased funding	*It is true that healthcare spending has increased in certain areas, but overall income remains the same. Spending also decreased in other areas because there is not so much surgery and emergencies of other pathologies.*
Material availability	*In my center there is no lack of other materials, but there is a lack of protective equipment; in this case not because of the crisis, but because of supply challenges.*
Personal protective equipment	*The PPEs were either donated to obtained by our center; not provided by managers.*
Influenza outbreak	*I remember during the first A-influenza we were equipped with the “duck beak” masks. Face shields were available for everyone, and there was plenty of Tamiflu medication.*

## Data Availability

The data presented in this study are available on request from the corresponding author. The data are not publicly available due to privacy restrictions.

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
