# Peer review of "Austerity Measures and Underfunding of the Spanish Health System during the COVID-19 Pandemic—Perception of Healthcare Staff in Spain"

_ijerph, 2023, doi:10.3390/ijerph20032594_

Round 1

Reviewer 1 Report

First, I would like to congratulate the authors for this very interesting work, and touch on such an interesting subject as the influence of resources and the perception of health workers during the pandemic.

Next, I suggest some changes and clarifications to improve your work.

-      In the introduction, some specific data on the economic deterioration and its repercussions on the health system are missing. Justification of this austerity and financial imbalance.

-      The study design is not clear, it must be defined and structured. If it is a qualitative study, the theoretical-methodological orientation from which the qualitative analysis is carried out is not specified, something essential for the analysis of the data and all the study.

-      Measurement instrument: the initials of the HLQ are not explained, it is intuited that it refers to the Health and Work Questionnaire. If this instrument has been used, this instrument and its psychometric properties should be described. It is commented that changes have been made, but no details are given about them and how they guaranteed the validity of the results if they modified the instrument.

-      The codes, dimensions and categories are not clear, if there was triangulation of the data and how it was carried out. It is also not clear how the data saturation was reached. The methodology should be reviewed as a whole.

-      The results are poor, you might analyze amount of information and it`s not in the paper . It is recommended a table with the most relevant categories and verbatims and expand the results. Also, a graph of how the analysis of the data has been carried out and the interconnections that they have been able to find.

-      The discussion is poor and does not compare the results with other studies.

-      In the bibliography, there are many studies in this regard from 2022, and its most current references are from 2021.

I hope these contributions are useful.

Reviewer 2 Report

Thank you for such important work. This topic is relevant to most Western countries, including the US. I would love to see the follow up work to see if preparedness for emergency response has been improved since COVID-19 and whether we learned the lesson.

Reviewer 3 Report

The article tries to shed light on a topic on which it will be necessary to meditate in depth, how the reduction of public resources in healthcare has affected the ability to respond to the pandemic.

The authors recall (lines 46 and following) how governments have allocated resources to deal with the crisis. it would be to evaluate and could be a topic for a subsequent development of the studies, whether emergency resources have improved the state of public health, or have not introduced further imbalances, for example by increasing the response capacity to acute problems but reducing that of chronic diseases.

This study was performed on a purposive sample of professionals, with a qualitative approach.

The interview guide questionnaire is well structured. The authors clearly report the methodology.

To correctly frame the results of their survey, and increase the generalizability of findings, they could refer to what emerged from the epidemiological surveys of healthcare workers, who in the first phase of the pandemic reported a shortage of protective equipment. Subsequently, the same workers reported problems deriving from the conditions in which critically ill patients were treated, in particular the isolation of patients from their relatives and the limitation of contact with health personnel. The isolation was also detrimental to the mental health of the doctors. These experiences were reported in Italy [see: Magnavita N, Soave PM, Antonelli M. Treating Anti-Vax Patients, a New Occupational Stressor—Data from the 4th Wave of the Prospective Study of Intensivists and COVID-19 (PSIC). Int. J. Environ. Res. Public Health 2022, 19, 5889. https://doi.org/10.3390/ijerph19105889 and the other articles of the same study]. The authors will also be able to find articles from other countries which confirm that the observations they have collected can be generalized. [see Waris Nawaz M, Imtiaz S, Kausar E. Self-care of Frontline Health Care Workers: During COVID-19 Pandemic. Psychiatr Danub. 2020 Autumn-Winter;32(3-4):557-562. doi: 10.24869/psyd.2020.557.

Luo M, Guo L, Yu M, Jiang W, Wang H. The psychological and mental impact of coronavirus disease 2019 (COVID-19) on medical staff and general public - A systematic review and meta-analysis. Psychiatry Res. 2020 Sep;291:113190. doi: 10.1016/j.psychres.2020.113190.
Cabarkapa S, Nadjidai SE, Murgier J, Ng CH. The psychological impact of COVID-19 and other viral epidemics on frontline healthcare workers and ways to address it: A rapid systematic review. Brain Behav Immun Health. 2020 Oct;8:100144. doi: 10.1016/j.bbih.2020.100144.].

Round 2

Reviewer 1 Report

Congratulations!!!